# 'I've Always Fought a Little against the Tide to Get Where I Want to Be'—Construction of Women's Embodied Subjectivity in the Contested Terrain of High-Level Karate

Fabiana Cristina Turelli [1,2,*], Alexandre Fernandez Vaz [3] and David Kirk [4,5]

1 Faculty of Kinesiology and Recreation Management, University of Manitoba, Winnipeg, MB R3T 2N2, Canada
2 Universidad Autónoma de Madrid, 28049 Madrid, Spain
3 Departamento de Estudos Especializados em Educação, Universidade Federal de Santa Catarina, Florianópolis 88040-900, Brazil; alexfvaz@uol.com.br
4 School of Education, University of Strathclyde, Glasgow G1 1XQ, UK; david.kirk@strath.ac.uk
5 University of Queensland, St. Lucia, Brisbane, QLD 4072, Australia
* Correspondence: fabiana.turelli@umanitoba.ca

**Abstract:** Karate can be both a martial art and a combat sport. Male and female karate athletes attended the Tokyo Olympic Games 2020 (2021). Elite sport often portrays female athletes through the sexualization of their bodies, while the martial environment leaves them open to accusations of masculinization. In the process of constructing themselves as fighters, *karateka* women do produce new ways of performing femininities and masculinities, which is a hard-work process of negotiations, leading them to the construction of a particular *habitus* strictly linked to their performativity within the environment. They take part in a contested terrain that mixes several elements that are often contrasting. In this article, we aim to present factors identified with the women athletes of the Spanish Olympic karate team that affect the construction of their embodied subjectivities. We focus on two main topics, authenticity as the real deal to belonging, and a possible gendered *habitus* struggling with the achievement of the condition of a warrior. We carried out an ethnographic study with the Spanish Olympic karate squad supported by autoethnographic elements from the first author. We focus here on the data from double interviews with 14 women athletes and their four male coaches. Embodied subjectivity as a process of subject construction to disrupt objectification and forms of othering showed to be a challenge, a complex task, and embedded in contradictions. Karate women's embodied subjectivities are built in the transit between resisting and giving in. Despite several difficulties, through awareness and reflection on limitations, *karateka* may occupy their place as subjects, exerting agency, feeling empowered, and fighting consciously against the naturalized '*tide*'.

**Keywords:** subjectivities; embodiment; negotiations; belonging; gender

## 1. Introduction

We know that women's place in elite sport is problematic for a number of reasons. Female athletes face diverse stereotyping to reach a position in the contested terrain of sports (Jackson and Scherer 2013), especially in environments understood as masculine, such as the *karateka*[1] world. By mixing a combat sport and a martial art, karate can host several patriarchal features that challenge women's determination to become fighters, and perhaps even more so for Olympic *karateka* athletes. In the context of sport as a male preserve (Theberge 1985), elite sport often portrays female athletes through the sexualization of their bodies, while the martial environment leaves them open to accusations of masculinization, which is linked to lesbian performativity (Bennett et al. 2017; Butler 1990). In the process of constructing themselves as fighters, *karateka* women do produce new ways of performing femininities and masculinities (Channon and Phipps 2017; Edwards et al. 2021; Maor 2018). However, this is not a process free from consequences, in relation to

peers and to oneself. The construction of a female *karateka* embodied subjectivity (Foucault 2016; Merleau-Ponty 2005) is demanding on women, requiring them to negotiate the construction of a particular *habitus* (Bourdieu 1990; Bourdieu and Wacquant 1992) that is strictly linked to their performativity within the environment.

The lack of recognition of women's capabilities coupled with the inferiorization of their athletic performance is already known in sports that are understood to be hyper-masculine, such as football, rugby, and combat sports (e.g., Mierzwinski et al. 2014; Turelli et al. 2022). This is so to the extent that if, on the one hand, women entering male-dominated environments can achieve levels of excellence and feel empowered, on the other hand, they need to deal with sanction and control (Edwards et al. 2021). Not unlike several other sports at the Olympic setting, karate retains and reproduces gender normativity. Following such an approach, studies taking into account male performances in karate have been carried out (for example, Alinaghipour et al. 2020), though not from a sociological perspective. As such, on an unproblematized binary gender order[2], general female performativity in karate has not been extensively researched. There is a literature on women in combat sports—for example, boxing (Carlsson 2017; Tjonndal 2019), judo (Guérandel and Mennesson 2007), and mixed martial arts (Jakubowska et al. 2016; Mierzwinski et al. 2014)—in addition to sociological reviews of the literature on the topic (e.g., Channon and Jennings 2014; Channon and Matthews 2015; Follo 2012), and women's self-defence from a physical feminism point of view (McCaughey 1998).

Notwithstanding, the specific literature on women's karate is still limited. Studies by Guthrie (1995) and Maclean (2015, 2016) have approached the gender theme, though neither includes women elite athletes within the Olympic context[3]. In this regard, our study makes a contribution to the social sciences, gender, and female sport, specifically deepening knowledge in the female elite[4] *karateka* context, where female embodied subjectivity struggles in interaction with a martial *habitus*. There are several studies exploring the concept of *habitus* in martial arts and combat sports (MACS), with Wacquant (2002) being a pioneer (see also Spencer 2009, 2012). However, our research focused on women in high-level karate taking the opportunity of the exclusive time, to date, of karate in the Olympic Games, Tokyo 2020 (2021), and specifically looked at embodied subjectivity, with *habitus* working as one strongly present parallel concept. It is also relevant to say that we decided to work with the concept of embodied subjectivity rather than identity considering that the former is a broader concept due to its sociological roots, expanding the psychological perspective of the identity topic. Also, in addition to knowing about the identities that *karateka* women can represent through belonging to karate, we are more interested in knowing how they build their subjectivity in such a contested terrain to deal with its several issues. We are considering the definition of contested terrain of Jackson and Scherer (2013, pp. 888–89) 'as a site of struggle (...) involving key interest groups with varying resources and material interests, and competing ideas and beliefs'.

In seeking to understand this process, we carried out a study with the women's Spanish Olympic karate team. The experiences they have embodied through karate, having practiced it since childhood, shape their experiences of themselves and of the world. In this paper, our aim is to present the factors identified with the athletes that affect the construction of their embodied subjectivity in the contested terrain of karate as an Olympic sport. This is important to be known because, by breaking down the process they went through, we can develop strategies to work in favour of women's sport, helping to make it a place where the objectification of women athletes is undermined. It must be said that there are several factors that help in shaping such embodied subjectivity, like personal struggles against hunger to be kept in weight categories, or overcoming pain from training sessions and injuries, and suffering from the pressure to win. However, we are focusing here on two not-so-explored topics in the literature, as far as we know, that link belonging to authenticity as *the real deal*, and a produced martial gendered *habitus*, to an ideal of the warrior. We chose to focus on these topics because, although complex, they are new to the

body of literature specific to women's karate, and present the potential of innovation for the area of women's combat sports.

In presenting this process of subject construction within karate, we first look at embodied subjectivity, providing an overview of this concept. Next, we briefly approach the quest to own a place and find comfort in belonging to MACS. The issue of belonging is present in sport in general, often being linked to a sense of safety; in karate, though, it can become a challenge enforced by the hierarchical system of belts as a stimulator path to be followed in order to integrate levels of mastery. The pursuit of a sense of belonging, then, highly affects how practitioners mould themselves to the environment. After addressing such a concept, we explain our methodological choices, considering that this paper relates to the central part of a broader ethnography. Finally, we present the section of findings and discussion, in which the women elite athletes express what it means for them to perform as *karateka* with characteristics that *define* them, helping to understand how embodied subjectivity is being constructed in the *dojo*[5] as a *habitus*-sharing environment, and on the competitive *tatami*[6]. The process they follow is not the same as that for men, showing itself to be a complex process, contradictorily guided by (hetero)normativity and a traditional martial pedagogy that places everyone in strict binary positions. To understand this multifaceted process, we add to the discussion the idea of a specific martial gendered *habitus* that seems to be present in this heteronormative gender binary environment. Both processes are connected, but we intentionally start with the topic of authenticity as we want to reflect its subordination to the gendered *habitus*. We would create a different impression if presenting the gendered *habitus* first, given that such an order of presentation could be read as a liberation process from impositions that the women face in the environment. Unfortunately, this is not what happens for now and we want to make it clear.

### 1.1. Women's Embodied Subjectivity

Embodied subjectivity is a dynamic concept that refers to the processes of the production of oneself (Foucault 2016) in an individuality sense, leading the person to become an embodied subject able to experience empowerment. A person's embodied subjectivity is built in the encounter of their lived experiences that capture the world through sensory perceptions with the reflexive processes ingrained in such living (Merleau-Ponty 2005). In this situation, karate could work as a technology of the self (Foucault 2016) supporting the process of empowerment, as it is reported in studies carried out in other combat sports (e.g., Maor 2018; see also Pedrini and Jennings 2021). However, high-level sport entails the risk, inevitably, of a view of the body-as-machine (Vaz 2001), while women athletes run the risk of being sexualized (see Toffoletti et al. 2018), with both processes feeding their objectification. A reflection on lived embodied experiences, therefore, is required as a component of self-cultivation (Foucault 2016), taking into account that we have/are selves that overlap in constituting our subjectivity.

Even though feminism has undeniably led to changes in the social and historical condition of women in many societies, secondary roles remain prevalent for them (Roth and Basow 2004). Patriarchal structures are organized and reorganized to maintain the hegemonic order (Connell 1995; Williams 1977). With that, women *karateka* may disrupt the gender order to some extent; however, in so doing, they face various adversities, be it in the general social context, in the sports world, or in the very traditional setting of a martial art.

We adopt Butler's (1990) concept of performativity to express our understanding of gendered embodied subjectivity. In performativity, repetition plays a central role, leading to reposition, so there is fluidity, and nothing is fixed. In having karate as a central element of life, the women athletes in this study *embodied* karate, although in different proportions, but enough to shape their performances, often in searching for authenticity as a way to feel and be seen as genuine through belonging to an environment that challenges their very situation as women (Young 1980; for an updated version, see Young 1998). They seem to find such authenticity in performing as *karateka* women, which is read as a genuine performance and is therefore more valued than something that could be considered a theatrical performance;

but this comes at a cost. They need to claim a position among fighters, belonging to the *karateka* subculture, which requires a set of negotiations, and this cost is increased further by reaching the Olympic scene, which makes karate more public and athletes more visible. To belong to the very small circle of *karateka* that could experience Olympic karate, or to be close to that by being part of an Olympic squad, positioned the researched athletes in a select place. Considering this, next, we briefly explain some elements of the process of belonging to an elite sport with a strong martial art background, which asks for surrender and adaptation to the local *habitus*.

### 1.2. Finding Your Place—The Comfort of Belonging through Habitus Embracement

In his work about techniques of the body, Mauss (1973) addresses the theme of imitation, which is a constant in sports contexts, with new practitioners becoming *mirrors* in the attempt to assimilate with the environment and resemble its respected members. When embodying an environment's *habitus*, a concept first mentioned by Mauss (1973) and developed as we use it here by Bourdieu (1990), the person forges themselves and becomes a constituent part of the group, which also accepts and incorporates the person (Mauss 1973). Then, the process of belonging is elaborated in a dynamic two-way street that is constantly being remade.

According to Spencer (2009), *habitus* can be conceptualized as an acquired ability and faculty, and is in place when the embodiment of body techniques happen. From this perspective, the embodiment of *habitus* refers to the habits or corporeal schemas described by Mauss (1973), notwithstanding, in expanding the focus of the somatic to an embodied subjectivity, and we recall Foucault (2016) on his technologies of the self. Karate does not detach from its background of martial arts even when becoming an Olympic combat sport. It retains invented or selected martial traditions (Hobsbawm 1983; Williams 1977) and philosophies that affect (Green 2011) athletes, as much as the general social and cultural learning process (Bourdieu 1990), setting up power dynamics.

In order to belong to elite-level karate, the athletes interviewed need to differentiate from amateur practitioners or traditional martial artists. At other times, however, the same high-level *karateka* recapture the roots that keep them connected to the martial art, perhaps romanticizing the traditional warrior's fraternity or sorority. There could be, therefore, a beneficial double-belonging, or yet a messy feeling of betrayal, to one of the scenarios, putting in evidence aspects of the contested terrain that sports karate at the same time takes part in and constitutes. Now we move on to presenting the methods employed in this study.

## 2. Methodology

This study draws from a larger (auto)ethnographic project led by the first author. The original plan, which we had already begun to enact during 2019 and early 2020, was to conduct an ethnography of the Spanish women's Olympic karate squad, in training and in competitions, following them to the Tokyo Olympic Games in August 2020. By the time of the arrival of the SARS-CoV-2 pandemic globally around February/March 2020, the first author had observed training sessions with the *karateka* and had begun to establish some relationships with them and their coaches. The pandemic changed our plan since training was disrupted and the Olympic Games themselves postponed until August 2021. To accommodate the global lockdown, we needed to adapt the initial research design and procedures. The pandemic, therefore, forced us to change methods and make compromises.

We decided to use some information from the first author's lived experiences[7] as a karate black belt and long-term amateur athlete in an autoethnographic approach (Landi 2018; Sparkes 2020), matched with data generated through online interviews with the elite athletes and their coaches, and video analysis of their competitive styles of fighting. For this paper, our analysis focuses mainly on data from one-to-one interviews with the women's karate squad and their coaches relating to the research question "what factors affect the construction of female *karateka* embodied subjectivities in the Spanish Olympic

karate team". We added extracts of the autoethnography where it is of more relevance. Our work was grounded in a social constructivist approach, supported by feminist studies of critical perspective (see Olive and Thorpe 2018)[8]. The theoretical framework added to the autoethnographic process led us to a dialogical and reflexive analysis of the interviews of the elite *karateka*, acknowledging a mutual affect as part of the process (Pavlidis 2013).

### 2.1. Participants

Our sample was composed of 14 women elite athletes, with ages ranging from 19 to 33, and who had practised karate for 13–26 years, as well as four men coaches, ages 33–62, with participation in karate ranging between 17 and 47 years. All participants were Spanish nationals, though from different regions of Spain. The athletes possessed between first and fourth *dan*[9], while the coaches achieved second to eighth *dan*. Among the athletes, ten practised *kumite*[10], and four practised *kata*[11].

### 2.2. Data Generation

We adopted an open-ended semi-structured interview (Hammer and Wildavsky 1990). We conducted two interviews with each participant. The first interview focused on general issues in the *karateka* environment and the second more tightly focused on the research topic. Over three months, we conducted 38 interviews, including a pilot between June and September 2020. The interviews lasted an average of 1 h 10 min, totalling more than 44 h of recordings. The co-authors of this article acted as guarantors of trustworthiness in the data analysis process, in a triangulation of scholar experts (Guba 1983) in the sociology of sport.

### 2.3. Data Analysis and Ethical Considerations

The data analysis began in early December 2020. The first action was transcribing the interviews, followed by coding them (Charmaz and Thornberg 2021), along with the translation process. We had a previous category system which was restructured based on real data obtained. From February to June 2021, the findings and discussion were written up. We followed all the procedures recommended by the Ethics Committee of the Autonomous University of Madrid to carry out the research, which considered that the project met its ethical requirements and approved the study in 2019, under approval number CEI-102-1930.

In the next pages we report the findings and discuss them in relation to athletes' perceptions of the construction of their embodied subjectivities that were influenced by their long-time belonging to karate and the gendered *habitus* verified with them. We note that some regions of Spain are characterized by taking strong feminist positions, which confronts with the extremely religious heritage that the people hold, affecting the social situation of the population. These embodied social features mix up with the martial *habitus* that the fighters find in karate, producing something of the same singular to Spanish karate, and plural, considering that Olympic karate is about a global movement. For more details on Spanish karate from a foreign point of view, please see Turelli (2022).

## 3. Findings and Discussion

We present findings that showed paths taken by women athletes to construct their embodied subjectivities in elite karate. We found athletes' concerns for authenticity, referring to being 'the real deal', as a genuine way of performing, and the relation of this to belonging, as well as a gendered *habitus* in the quest for warrior status. Both topics are supported by contradictions found in the contested terrains due to the 'power struggles over resources (...) and ideological and moral/ethical beliefs' (Jackson and Scherer 2013, p. 889). Yet belonging is about taking part in a current environment; it generates the feeling of authenticity when the achievement of such a condition is obtained through relevant challenges and produces a sense of exclusivity. A gendered *habitus* that places people according to traditional and gender binary stereotypes becomes controversial when the ideal model proposed is fixed and the same for all. Such contradictions and contrasting

points add to the complexity and richness of the subjectivity that is being constructed by women due to its need for negotiations, since female space in the considered-masculine environment is still not guaranteed, but a continuous struggle.

*3.1. 'We Are Authentic'—What It Means to Belong to a Karateka Group*

Sometimes women *karateka* consider themselves athletes, sportspeople like any other, dealing with issues shared among the sports community; at other times, they expose the view that karate gives to its practitioners some special characteristics, providing them with a sense of exclusivity and originality outside of what could be considered commonplace, and feeling superior in relation to other people and sports. They address such a sense as an experience of authenticity, which can be understood as being 'the real deal', something genuine with the value of legitimacy, distanced from falsehoods. Achieving a high-level position is something that places people in a restricted circle, as Diana explained:

> *It's high-level sport. (. . .) It is a pyramid, so at the top of the pyramid not everyone fits.* (Diana, Interview13(2), 13 August 2020)

To perform as a woman in such an environment is something that can confer status due to the peculiarities of the *karateka* field. A woman who fights, despite all the difficulties she faces in performing it, can find in such a condition a place of authenticity, meaning that she differentiates herself from other people, feeling exclusive, brave, maybe even special, in a sport that is yet not widely embraced by women. Then, there is the combination of belonging with feeling like a genuine *karateka*. This is intimately known by the first author, who used her personal experiences in karate to decode some reported information through interviews with the elite athletes, as pointed out in the methods. Remembering her path, she wrote

> *You take part in the context and become part of the group after passing the challenges presented. (. . .) And I would add that even though all these things come at a price, being the only girl training among boys 25 years ago in a countryside town guaranteed a remarkable exclusivity. In other words, all my friends (girls) were surprised that I was doing what none of them had the courage to do, even if I invited them to participate. So I confess that I felt authentic.* (Fabiana, autoethnographic text, in Turelli 2022)

The search for this place to feel genuine often leads people to attribute a special status to what they dedicate themselves to. Although many of the women consider themselves to belong to a generic community of *athletes* on the one hand, on the other hand most have the view that performing as *karateka* represents the insertion of exclusive values in their lives. They reveal their belief in being part of an inner circle of martial artists as the next testimonies attest:

> *I think that karate moulds your character and way of being, and we do have specific characteristics that maybe you don't have them with other sports.* (Hera, Interview21(2), 27 August 2020)

> *Inside of us we can feel something that makes us different from others, I don't know exactly what it is. The way you take things, respect.* (Artemis, Interview26(2), 7 September 2020)

> *I think that karate does help me to be an educated, respectful person, that opens up paths that other sports may not.* (Minerva, Interview12(2), 12 August 2020)

> *I sincerely believe that karateka have a different mind than any athlete or person. I believe that we have been so governed by discipline, rectitude (. . .), and that does not apply in all sports but within karate it does.* (Venus, Interview22(2), 2 September 2020)

*Karateka* cite values that shape character, as advocated by Funakoshi, the accepted founder of karate, and mention concepts related for them to the distinctive and somewhat special way of performing as *karateka*. They are elite sportspeople like other athletes, but also consider themselves to belong to something differentiated, in a superior manner.

Donnelly (2006, p. 220) notes that the position of 'us against them', to some extent found here, clarifies 'differences between core subcultural participants and the people they define as other'. Since performing as a fighter involves taking some risks that not everybody is willing to take, this is linked within the environment to courage, a valued feature among karate people that adds to the authenticity they seek.

To verify to what extent belonging to karate affects athletes' embodied subjectivities, we also asked them 'who are you?'. The question showed itself to be difficult to answer. We share two comments:

*Complicated this question! (...) When I fight at my best, I am authentic. Maybe I don't have a spark, maybe my matches don't seem super attractive, super entertaining. But I do my job.* (Diana, Interview13(2), 13 August 2020)

*Well, I think I'm a fighter. In many ways. Because it has cost me a lot to get to where I am now. (...) So I've always fought a little against the tide to get where I want to be.* (Minerva, Interview12(2), 12 August 2020)

The authenticity that Diana sees in herself, as do other athletes, contains contrasting elements. It seems that she poses her authenticity as a way of genuine performance, against what is expected of elite athletes in terms of spectacle, of entertainment. However, if she wins, by doing her job, she is worthy of remaining in the squad. Minerva emphasizes that she is not following the tide as well; the place she wants is not a common place, but a place that requires a fight and effort to be conquered. Then, both athletes believe themselves to be adopting positions, to some extent, with an extra value, maybe with differentials of what can be considered as a community of athletes. Their embodied subjectivities, therefore, are produced based on the feeling of being the real deal in the very effort to belong to a place that challenges, but also reinforces, their performances, and on the sense of self-worth as warriors, which will be explored in the next topic.

### 3.2. 'We Are Warriors'—The Pursuit of Characteristics of Ideal Karateka and the Presence of a Gendered Habitus

The search for acceptance into and belonging to a group justifies the embodiment of the local *habitus*. However, there seems to be a contrasting gendered *habitus* that makes subjectivity construction a complex task for women. This concept may present itself as an extra reason for the way women conform, while they resist diverse issues within the terrain of karate.

The construction of embodied subjectivity is affected by the strong value that is attributed to a warriors' performance. There is a pedagogy behind the actions taken in martial arts that promote a somewhat romanticized view of warriors and masters. In a contrasting position with a combat sport fighter, a warrior would usually be a male martial artist with a moral code, like a *samurai*, following a powerful and respected tradition (Hobsbawm 1983; Williams 1977). Indeed, Cynarsky et al. (2012) describe such a pedagogy as a way of moulding the character for the practitioner to achieve an elevated moral status. It must be noted that the education in martial arts is usually under a strong gender binary and straight view of sexuality; it is heteronormative (see Turelli et al. 2022). Therefore, women performing karate often seek to fit into the binary hierarchical order. This order means, in general terms, that men under the martial pedagogy are authorized to develop high standards of fighting, while women pursue this ideal model of warrior, but are rarely viewed (by mostly male *sensei*[12]) as reaching the criteria for doing so.

Then, even though karate has become an Olympic combat sport, the place women hold within it is mired in complexities. Unlike in mixed martial arts, where 'A fighter becomes a mixed martial artist in the experience of being in the ring' (Spencer 2009, p. 136), far more is required for women karate fighters to become warriors. Women are seen as being authorized to take part in the male environment, as it is *naturally* (Young 1980, 1998) considered to be, and thus they are not seen as true fighters, but receive instead an inferior classification to men within the context (Turelli 2022). They have their fighting

styles compared to men's, even though, following traditional martial pedagogy, they neither receive equal education to men, nor is it based on equity, but on the wider-known traditional approach. Then, if they somewhat disrupt the normative gender order several times through pursuing the (male) ideal of a warrior for the context, they risk having their performativity stereotyped, falling into accusations of lesbian performativity, as Venus pointed out:

> *Many people think that, that we are* machungas *(lesbians).* (Venus, Interview22(2), 2 September 2020)

The assumption of lesbian performativity for participating in a male sport environment tends to be frequent and becomes a stereotype that women need to deal with, regardless of their sexual preferences. In such situations and contexts, women athletes react in ways that reinforce heteronormativity by exaggerating their femininity (e.g., Tajrobehkar 2016) as a way to deny that they identify as lesbians. It relates to embracing emphasized femininity (Connell and Messerschmidt 2005) through, e.g., using makeup, wearing tight, sexy clothes when not wearing a *gi*[13], wearing high heels, and so on. Venus still can extract the benefits from such a stereotype by converting it into a way of feeling genuine. She gives minor importance to the stereotype since she considers herself to be a 'real girl'. This, in turn, can be read as an embodiment of the traditional gender binary and straight pedagogy (Fitzpatrick and McGlashan 2016). Venus said:

> *I think we are authentic. Because I think we are the ideal girl that everyone thinks is a boy, but when they meet you, they really realize that you are a real girl.* (Venus, Interview22(2), 2 September 2020)

Then, women seek the given high standard of fighting, proving their value as warriors, worthy of belonging to the group, but also striving to portray themselves as 'real girls' that display emphasized femininity and do not identify as lesbians; another contrasting point, since girly women may be distanced from holding a supposed true warrior status. Thus, to belong to the environment, *karateka* women, particularly at this elite level, make a series of negotiations. At some points, they position themselves within this contested terrain as resisting forms of domination, and at other times they (need to) give in to keep their place in the team (Turelli 2022). They end up embodying the martial *habitus*; notwithstanding, it is to some extent adapted to a female version of warriors, not understood as 'true' warriors in the context, though given the supposed female *natural* condition (Young 1980, 1998) with the role of supporting men. Atena, an interviewed athlete under a pseudonym, provided an example of the negotiations she makes even when she holds power due to the second role she holds. She reported

> *Within the coaches' group, when we go to a championship I am a girl (coordinator of the squad at her province), I have to catch their attention. I must tell them what they are going to do, and they are reluctant, and it is "what is this girl going to tell me?" So what I'm trying to do is. . . like being neutral, trying to say things in a way that doesn't hurt. So if I have to scold him, even though I would yell at him, what I try is (. . .) "look, this can't happen again because. . . I know that anyone makes a mistake. . ." You try to handle a little better, because if the situation that I am the authority can produce rejection, if I was more hierarchical. . . (would be worse). A man who is in power, if you fail, he would throw you off. But I try to do it like a conversation, I call you aside, we talk. . .* (Atena, Interview14(2), 15 August 2020)

Female *karateka* seek to reach the ideal of a warrior's fighting style, accommodating this request with the situation of women that is viscerally embedded in them by society (Mason 2018) and by the traditional martial pedagogy itself. It is contradictory because they are asked to perform in line with the given ideal model, but it mixes up their social self-positioning. They want to achieve warrior status to find their own self-worth, and keep their 'real girl' status, positions that they continue to struggle with. In such a pursuit, they deal with elements that are demanding to their holistic health and wellbeing, elements

challenging men fighters as well, but that echo loudly among women since they are considered to be in a men's area. It seems that to perform at the same time as *real girls* and *real fighters* could give them *the real deal* of *karateka* women. They commented that

> *Karate is a sport considered more for boys, so for girls to be there they must have character, a strong personality.* (Vesta, Interview16(2), 18 August 2020)

> *I think that in the end a woman who gets into a sport that a priori is considered masculine, first, has a (strong) personality. And courage or will.* (Afrodite, Interview19(2), 24 August 2020)

These comments show the influence of the martial *habitus* and ways that athletes find to deal with it, making negotiations to conquer the position they want, as well as showing the impacts this can exert in their embodied subjectivity. In attesting they are warriors, they point out values that are highly considered as values they carry, and also highlight some characteristics that they could portray as an advantage regarding men. The athletes explained:

> *We use intelligence much more than boys. Boys are more physical, and for that reason they also score more points. Generally, the scores of the boys' fights are higher, I think they defend less, they go crazy.* (Minerva, Interview12(2), 12 August 2020)

> *Guys get hotter. (. . .) I don't want to say that we girls don't get stung, because we do it a lot, but we know how to do it in a different way than simply "well, now I get stung and hit and destroy you."* (Diana, Interview13(2), 13 August 2020)

Women define their characteristics as opposed to what would be viewed as the strong point in men, corroborating the current and normative gender binary view of men and women as well as the pedagogy under which they are being coached. It means, while they claim for equality, it is not possible to detach from the viscerally embodied social gender binary values.[14] This is why they emphasize the 'head' as a main characteristic of *karateka* women, a synonym of performing more rationally, intelligently and even strategically than men, who would be more impulsive and 'physical'.

However, athletes and their coaches present a contrasting view; coaches undeniably recognize men as physically superior and more skilled than women, but they also see men athletes with greater strategic capabilities. So, while women consider their own fighting style to be rational and calculated, coaches will point out that this as actually a deficit in females, again remarking the gendered *habitus* and placing women as inferior fighters. The tactical aspect is what characterizes good fighters for coaches, and what, according to them, *karateka* women lack. One coach stated

> *We find very good girls, with a lot of technique, raising their legs, very fast, but then at a tactical level. . . Today at a strategic level that does not help you if you are not able to deceive the opponent, make her fail. . .* (Coach Apolo, Interview33(2), 25 September 2020)

The coach refers to the level not reached by women in his view. Women want to prove their worth for fighting and that the shortcoming pointed out by coaches is mistaken, which especially involves them in working hard to reach a sense of belonging within the team, but also due to the internalization of the hierarchical figure and his gaze (Foucault 2009). However, their pursuit follows orientations received by the people who *deeply know* (and hold, like the true owners of) the martial art, who are the coaches. Thus, they internalize messages, try hard to attest their ability as warriors, and sustain inner battles against the beliefs of inferior performance that they supposedly present (see Turelli et al. 2022). Another athlete provided a testimony in this regard:

> *I have heard it in the coaching courses, "with the girls the tactics cannot be worked". Author: And why? Atena: Because they say we are unbearable, that you say to a girl "look, what you are going to do is. . .", imagine, "well, now I want you to dodge and do. . ." and that we start crying, that we have a very bad character, that I don't know*

*what. (. . .) I tell them "well, I'm not like that", and then they tell me "it's because you're a boy." (meaning that she performs strongly, has not so many sexual appeals. . .) That is the turn. The first time "you can't because women are unbearable", and the second time is "of course you do, but because you're a boy."* (Atena, Interview14(2), 15 August 2020)

In addition to the contested terrain that the contrasting comments of coach Apolo and Atena evidence, Atena provides a testimony that corroborates the point on the masculinization stigma of women fighters, leading back to Venus' quote presented earlier on lesbian performativity. Atena still shows how *karateka* feel the impacts of the binary martial pedagogy that positions them as poor fighters (Turelli et al. 2022). They are playing in the competitive sports world, which hinges on victories and defeats, the former usually leading to high status and the latter to the possibility of being humbled and ultimately ejected from the squad. Then, there can occur experiences that are read as dishonourable. We provide an example from our autoethnographic notes:

*I punched the senshu[15], and then I received the first kick to the head. (. . .) With the second kick in the head and the score at 1 × 6, I was already in my shame process, worried about several things external to the fight, and was not reasoning well, which made room for the third kick and the defeat by 1 × 9. This shame is absurdly terrible, especially in the moment, but also after and even now, years after the fight took place.* (Fabiana, autoethnographic text, in Turelli 2022)

Insisting on the gendered *habitus*, the internalization of judgments and opinions, and some allowance of being constantly evaluated by others (Foucault 2009), lead women to become their own regulators. Minerva provided an example of this:

*I am a self-demanding person and I don't need nobody to be behind me to try my best. I can do it alone.* (Minerva, Interview1(1), 29 June 2020)

To do this with oneself requires remarkable moral strength, regardless of the normalization and acceptance of the load of the judgment of others, to the point where it becomes embodied as self-demand. It can be read as self-discipline and praised in the sports world, but also as a disciplinary way of living (Foucault 2009), where the body needs to progress in a disembodied approach, like a machine (Vaz 2001). The entire process can end up being felt as heavy to carry, often linked to experiences of guilt for not achieving (self)imposed targets. Guilt is in a close relationship with shame, and is a form of self-punishment for thinking that the ideal fighting style is not achieved, even if perpetually pursued. Then there is the feeling of non-completeness and the *non-abled* fighter (Turelli et al. 2022), who lacks ability and is ultimately inferior. Some athletes commented on how they struggle with this issue:

*That's why I didn't feel good, because I threw extra rocks in my backpack that I didn't have to carry.* (Diana, Interview4(1), 21 July 2020)

*I do think "I'm not good for this, I'm never going to get it." Those kinds of thoughts, yes. . .* (Vesta, Interview8(1), 25 July 2020)

*I have a (inner) saboteur who tells me bad things and I tell myself it's a way to protect me, like when you make excuses, "I'm going to lose, I'm going to lose", like if you lose, you were already saying you were going to lose. (. . .) It's a constant fight.* (Artemis, Interview23(1), 3 September 2020)

This self-disciplinary approach and the burdens taken as self-punishment relate to the gendered *habitus* found among these women. The prominent place given to a specific posture that must be presented results in pressure, concern, and self-depreciation, contra-values embodied that lead women fighters to produce a sub-*habitus* within the martial *habitus*, generally spread among female *karateka*. This feeds a cycle of self-blaming for performing in modest ways, with restricted movements in space, for example, when it was

instilled in the education on the female body from infancy (Mason 2018; Young 1980, 1998). In this regard, Atena pointed out

> *I believe that if I'm a coach and I'm used to always receiving from others "you didn't do this well", and you are judging yourself harshly, I think that that makes women in general much more insecure than men. Because there is always a criticism of what you do, always. If you go very covered, you are a nun, if you go uncovered, you are a whore, if you paint yourself, why do you paint yourself, and if you don't paint yourself, you don't explore yourself. There are always comments on everything about you. I think it conditions you. (. . .) Many women say "and can I do that? And will I be able to. . .?" Because they have always put it in you that it's not your site, that you can do things wrong. (. . .) I think that it's not the same to go on a flat road than to go uphill. If you put me uphill, it makes it much more difficult for me to arrive than for you.* (Atena, Interview14(2), 15 August 2020)

We argue that female athletes pursue the archetypal ideal fighter that is pointed out by their coaches and the environment whilst navigating contradictions, and it strengthens this peculiar gendered fighter *habitus*. Women *karateka* under the traditional martial pedagogy education do not enjoy the same place than that designed for men. Notwithstanding, they keep stressing that they can perform as *true* warriors:

> *We are warriors. (. . .) I think warrior is a word that defines us quite well.* (Hera, Interview21(2), 27 August 2020)

> *We fight to the end, and that defines us, not giving anything up.* (Minerva, Interview12(2), 12 August 2020)

> *I would say that (we are) brave, moving forward, that nothing stops you, nobody.* (Ceres, Interview17(2), 23 August 2020)

Women want to prove their self-worth for the sport through the emphasis they place in their comments, and it must be noted that they are indeed overcoming limits and barriers both on the mat, for their space in the sport, and in life itself. The social construction of women counts on constricting their achievement of goals that are reflected in the sports world. Notwithstanding, they continue to fight for their place, and this very fight, even if full of challenges, is outlining and constructing subjectivities *enabled* to exercise agency over traditional structures and contested terrains, despite the rhythm it takes.

## 4. Conclusions

In this article, we aimed to present factors identified with the women athletes of the Spanish national squad that affect the construction of their embodied subjectivities in the contested terrain of karate as an Olympic sport. We focused on presenting that which seemed not-so-explored in the literature, and with the potential to visualise paths for women's sport. We presented two main topics, one relating to authenticity, as being the real deal and thus deserving to belong, and another about a gendered *habitus* struggling with the achievement of the condition of a warrior. All this is developed in an environment permeated both by a traditional martial pedagogy, which still follows heteronormative gender binary rules, and by the known sportive culture that objectifies women athletes. These elements are made up of combined, complex and contrasting points that mainly affect women in their process of constructing embodied subjectivity, a hard-work process.

Such contrasting points contribute to shaping subjectivities that deal with moments of insecurity and are due to the help in leading to the verified gendered *habitus*, which is therefore not a plenty confident *habitus*. The *Karateka* athletes that were researched take part in the elite sport; however, they keep attached to the traditional martial approach that karate has, which is rooted in a pedagogy that does not favour them. Since martial pedagogy is added to the sportive culture, women fighters struggle and try hard to belong to this contested terrain. Notwithstanding, the goal is worth it to them, because conquering a place within the environment provides an important feeling of differentiation from others,

finding in it a way to experience self-value. Having the possibility to feel 'special' becomes something intensively pursued when to feel relegated to a second level is commonplace.

Added to this is the quest to be viewed as *real fighters*, as recognized warriors. *Karateka* pursue the male model posed as the best way of fighting, even though it is not achieved by most male fighters. In seeking that, women face prejudices around their sexuality, and so they try to attest that they are *real women*, emphasizing their femininity and being compliant, at times at least, to figures within and general martial education and training that see them as 'others', not working with them equitably, but othering them. They are, from this perspective, not equals, but only allowed to play within the male terrain, pretending that they fight. Even though they fight amazingly, they could achieve higher levels if their sport was approached as specific sport, not as an 'imperfect inferior imitation' of sport displayed by men. Therefore, the comparisons they are subject to are untenable and unfair.

The sports world is a contested terrain of great social power and numerous power struggles, working with people in their entirety and acting on their subjectivities in the interaction of the practiced *habitus*, gendered embodiment, and embodied subjectivity. By knowing and better understanding the nature of athletes' embodied subjectivity, we may become more prepared for proposing changes in benefit of women's sport. Athletes face several difficulties to perform their sport; they embody the culture of the martial context through its traditional pedagogy and a remade *habitus*, and negotiate versions of themselves produced through others' eyes, resisting at times, and giving in on other occasions. It is undeniable that they need to adapt to some norms of *fitting in* to enter and remain in the team. On the other hand, they also defy or confront structures when fighting for aspects of their conception on authenticity, performing in genuine manners. But there are limits to this. They need to know or to learn how to negotiate and balance resistance with yielding to norms, being able to resist more when their results are better, maybe. Then, to perform authentically could be a way of resistance that is, in turn, guided by oscillating orientations between fighting like the ideal warrior and holding back, restricting, demure as the gendered *habitus* presupposes.

Considering this contradictory scenario, the fact that women stand for a place where they are so challenged shows they are able to occupy that space, which can be addressed as a process of deepening in and fighting for subjectivity construction, disrupting objectification. Through awareness and reflection on several limitations, a process that took place alongside the interviews, *karateka* may occupy their place as subjects, feeling empowered and fighting against *accepted* forms of othering. Perhaps this is the first *tide* to fight against; not an easy task, but rewarding. Karate women's embodied subjectivities are built on this contested terrain precisely in the transition between resisting and giving in, needing to find a sort of centre in this movement without destabilizing oneself. This presupposes a position of conscious struggle with power dynamics, a *fight* continually re-elaborated in a process of reflexivity, and in developing a critical and political stance. Therefore, in addition to a defying challenge, there is also an opportunity with potential to be explored in further studies.

**Author Contributions:** Conceptualization, F.C.T. and D.K.; methodology, F.C.T. and D.K.; validation, D.K. and A.F.V.; formal analysis, F.C.T.; investigation, F.C.T.; resources, F.C.T.; data curation, F.C.T.; writing—original draft preparation, F.C.T.; writing—review and editing, D.K.; supervision, D.K. and A.F.V. All authors have read and agreed to the published version of the manuscript.

**Funding:** This research received no external funding.

**Institutional Review Board Statement:** The study was approved by the Ethics Committee of Research of the Autonomous University of Madrid (protocol code CEI-102-1930, in 2019).

**Informed Consent Statement:** Informed consent was obtained from all subjects involved in the study.

**Data Availability Statement:** Due to privacy protection, data of informants are not provided.

**Conflicts of Interest:** The authors report there are no competing interest to declare.

## Notes

1      Karate practitioners or environment.

2      We would like to make a point here highlighting that our acknowledgement of the contested terrain (Jackson and Scherer 2013) of sports, and specifically the *karateka* environment, as spaces where the binary gender order prevails refers to itself as a fact. Undoubtedly, it is a fact that that increases the situation of the contested terrain, amplifying contradictions both in number and dimensions. However, in pointing out our awareness of the issue, we are not advocating for an undeliberated mix among all genders for competitions. We mean competitions and not recreational sport. By experiencing competitive sport, where specific criteria matter for performance, and where how to (fairly, we hope) exclude ends up being more emphasized than inclusion, we do think that binarism is an outdated issue that needs to be addressed in sport. Unfortunately, though, we do not have an easy solution to propose for that. At this point, we see disadvantages for both trans and cisgender people, the former mainly related to inclusion topics, and the last to (mainly cis women's) rights in sports, which have been achieved through a journey of struggles far from an end, given female sports' continuous comparison and devaluation in relation to male sport. Yet, specifically thinking about competitive combat sports, and again considering our embodied experience, mixing genders for fighting, even with the rules of sports, can lead to levels of danger in experiences higher than the habitual.

3      Karate debuted in Tokyo 2020 (2021), but is no longer included in Paris 2024.

4      Elite athletes mean here *karateka* that are not amateur and integrate a national squad. They are not necessarily professional though, studying or working in other areas.

5      Martial practice location.

6      Area of practice.

7      Since ethnographies delve on people's experience, they are often linked to the phenomenological approach (Spencer 2009). We are aware of this but have chosen a different path to serve our study's object and go forward to what Fullagar (2017) named as comfortable known qualitative research.

8      For an evolving perspective from us, see also Bargetz and Sanos (2020).

9      Grades after black belt.

10      Fight itself, structured by weight categories.

11      Fight against an imaginary opponent, a *choreography* of martial blows.

12      The graduated teacher.

13      Abbreviation used to the karate uniform (*karategi*).

14      There is a sort of branch of feminism like the one in defence of equity, with which we identify, and that at several times athletes are in accordance with. However, here we consider a contradiction to be in place due to common sense, in Gramscian terms.

15      *Advantage* obtained by scoring the first point.

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
