# Peer review of "‘I’ve Always Fought a Little against the Tide to Get Where I Want to Be’—Construction of Women’s Embodied Subjectivity in the Contested Terrain of High-Level Karate"

_socsci, doi:10.3390/socsci12100538_

Round 1
Reviewer 1 Report
I think that this paper has enormous potential. It details a field (Bourdieu) that offers a number of interesting contrasts and juxtapositions, many of which were described by the authors and some that I will add:
1) The gendered habitus of karate, emanating from both its history and its sociology, which was explained well by the authors and had some support from the participant narratives and the autoethnographic note (p.6 line 266ff). The contrast between coaching comments and athlete comments, especially around rationality (second wave feminism would explore this further) was contested terrain. I also liked the long quote from Atena on p.8 lines 389-398, who was trying to straddle the gap between coaching and gender.
2) Overlaying this was the juxtaposition of Olympic Sports (winning is everything) with martial arts (repetition, conformity, obedience to masters). I think that the authors could have made more of this- it was mentioned early but not particularly developed.
3) Add in the authentic warrior, and the different ways that women and coaches viewed both authenticity/realness and warriors. Particularly interesting was the recognition by the women fighters that their warrior status is more authentic because they must 'go uphill' (p. 11 line 527). This was well covered by the authors, but could be improved by either/both greater investigation of the literature on feminism and martial arts subjectification.
4) And finally, the idea of the public presentation of the women fighter against the larger gendered norms of the society (and of much elite sport media coverage). The heterosexual norms were only outlined in one short participant narrative by Venus- consider whether the sidetrack onto explaining this norm interferes with the more general narrative flow about gendered norms.
5) One extra that could be looked at, if participants brought it up at all, was any differences in narratives between athletes that practiced kumite and athletes that practiced kata. Is there a gendered difference, related to authenticity and warrior status, related to these two areas.
My problem is that I think that the paper could have developed the conceptual foundations for explaining and analysing this contested terrain with a much deeper analysis of both martial embodiment and feminist analysis.
For the first, I would recommend the following literature (although there is more recent stuff):
McCaughey, M. (1997). Real knockouts: The physical feminism of women’s self-defence. New York, NY: New York City Press. (I think that McCaughey's work would really help you)
McCaughey, M. (1998). The fighting spirit: Women’s self-defense training and the discourse of sexed embodiment. Gender and Society, 12(3), 277–300.
Young, I. M. 1998. “Situated Bodies: Throwing Like a Girl.” In Body and Flesh: A Philosophical Reader, edited by D. Welton, 259–273. Oxford: Blackwell. (I specifically added the later Young article because it addresses her earlier article after 25 years of Title IX in the US. I presume there is a similar expansion of sporting opportunities for women in Spain over this period. In this later paper, Young explains the development of an embodiment that is significantly different to that described in Throwing Like A Girl. For Young, embodiment of women is learnt and disciplined, and resistance is possible and growing.)
For the second, and without wishing to influence you in ways that you do not want to go, consider the current literature on feminist new materialism, which explains the ways that the body interacts with material ‘reality’ in the spaces it occupies. That is a massive simplification, but I found the following literature useful:
Bargetz, B. and Sanos, S. 2020. Feminist matters, critique and the future of the political. Feminist Theory, 21 (4), 501-516. 10.1177/1464700120967311 (Uses Karen Barad on new feminist materialism)
(2017) Post-qualitative inquiry and the new materialist turn: implications for sport, health and physical culture research, Qualitative Research in Sport, Exercise and Health, 9:2, 247-257, DOI: 10.1080/2159676X.2016.1273896 (You mention Simone Fullager’s article in note 7, but her paper offers a great framework for your paper.)
Toffoletti, K., J. Francombe-Webb, and H. Thorpe. 2018. “Femininities, Sport and Physical Culture in Postfeminist, Neoliberal Times.” In New Sporting Femininities: Embodied Politics in Postfeminist Times, edited by K. Toffoletti, H. Thorpe, and J. Francombe-Webb, 1–19. London: Palgrave Macmillan. (Any of Toffoletti’s, Thorpe’s, Olive’s or Pavliis’ recent work would be useful)
Olive, R., and H. Thorpe. 2018. “Feminist Ethnography and Physical Culture: Towards Reflexive, Political and Collaborative Methods.” In Physical Culture, Ethnography and the Body: Theory, Method and Praxis, edited by M. Giardina and M. Donnelly, 114–128. London: Routledge.
Pavlidis, A. 2013. “Writing Resistance in Roller Derby: Making the Case for Auto/Ethnographic Writing in Feminist Leisure Research.” Journal of Leisure Research 45 (5): 661–676. 10.18666/jlr-2013-v45-i5-4368.
C., 2016. Getting lost as a way of knowing: the art of boxing within Shape Your Life. Qualitative research in sport, exercise and health, 8 (5), 472- 486. 10.1080/2159676X.2016.1211170
If you don’t want to go there, then I think that you need to dive a bit deeper into the literature on the patriarchal context of sport, and the various methods that reinforce the hierarchy that places men over women. You certainly covered this idea when talking about warriors, but this also needed a deeper conceptualization.
One final comment on content- I’d remove footnote 2. It does not add to your paper and potentially detracts from the potential for women to find resistant subjectivities through karate. My very limited understanding of karate is that women often spar with men in training. Additionally, the rigorous educational and accreditation systems (belts) along with the rules of engagement, would allow for the theoretical possibility of mixing the genders in combat. I acknowledge your final point in this note, but someone like McCaughey would probably argue differently. But this may be especially so when you consider what your research said about authenticity and warriors.
One specific comment- delete the sentence (or at least the reference) on p.4 line 186-187 to Bowman. Bowman's paper is about how Covid impacted on his relationship with martial arts, and not about how Covid impacted on research design.
I want to acknowledge the efforts of the author(s) to translate participant comments across languages. I have never had to endure this, and it must be difficult.
My comments will be mostly critical (that is the nature of a review unfortunately), but I also hope that they are constructive.
I found that the authors used a number of complex sentences where the understanding of the manuscript would be improved with more and shorter sentences and paragraphs. A strong edit to remove wordiness where it is distracting is needed. I suspect that there were two reasons for this:
A) An endeavour to squeeze this research into the idea of ‘contested terrains’ by mentioning this concept as often as possible- In my view this was both unnecessary and distracting.
e.g. p. 2 line 76-79- I don’t see the need to define contested terrain here. If you want to do so, then shorten to ‘a site of struggle…involving key interest groups with varying resources and material interests.’
B) A desire to constantly link back to existing research literature. This again became distracting in trying to follow the narrative flow of the paper.
e.g. p.4 lines 160-165- You could easily cut out the references to Hobsbawn and Williams, and probably Bourdieu and Green, and have a much tighter sentence structure which emphasizes the main point such as:
'The history of, and philosophies underpinning, martial arts create a field where the teachings of a school or master set up hierarchical power relations.'
These are two examples of many in the paper where you could lose words whilst producing a stronger narrative flow.
Finally, you need to redo the reference list to use a consistent style.
Author Response
Please see below my replies in green. Thank you.
Reply to Reviewers
Reviewer 1
I think that this paper has enormous potential. It details a field (Bourdieu) that offers a number of interesting contrasts and juxtapositions, many of which were described by the authors and some that I will add:
1) The gendered habitus of karate, emanating from both its history and its sociology, which was explained well by the authors and had some support from the participant narratives and the autoethnographic note (p.6 line 266ff). The contrast between coaching comments and athlete comments, especially around rationality (second wave feminism would explore this further) was contested terrain. I also liked the long quote from Atena on p.8 lines 389-398, who was trying to straddle the gap between coaching and gender.
2) Overlaying this was the juxtaposition of Olympic Sports (winning is everything) with martial arts (repetition, conformity, obedience to masters). I think that the authors could have made more of this- it was mentioned early but not particularly developed.
3) Add in the authentic warrior, and the different ways that women and coaches viewed both authenticity/realness and warriors. Particularly interesting was the recognition by the women fighters that their warrior status is more authentic because they must 'go uphill' (p. 11 line 527). This was well covered by the authors, but could be improved by either/both greater investigation of the literature on feminism and martial arts subjectification.
4) And finally, the idea of the public presentation of the women fighter against the larger gendered norms of the society (and of much elite sport media coverage). The heterosexual norms were only outlined in one short participant narrative by Venus- consider whether the sidetrack onto explaining this norm interferes with the more general narrative flow about gendered norms.
5) One extra that could be looked at, if participants brought it up at all, was any differences in narratives between athletes that practiced kumite and athletes that practiced kata. Is there a gendered difference, related to authenticity and warrior status, related to these two areas.
Thank you very much for your points. We have reviewed the entire manuscript seeking to improve it while following your observations. We also tried to respect the word limit given to us. Specifically for point 5, we have a book chapter in review that addresses hierarchies in karate, with a hierarchy of aesthetics (for kata practitioners) and another of bravery (for kumite practitioners) being approached. Thus, we do not enter this point in this article. And for point 2, we focus on this issue in a paper under review about gendered karate in neoliberal society, therefore not using the word limit for this here. For point 4, we considered deleting Venus’ quote but gave up on that because her quote serves as a reinforcement of the traditional martial pedagogy placing (cis) women as “real women” who supposedly should identify as heterosexual.
My problem is that I think that the paper could have developed the conceptual foundations for explaining and analysing this contested terrain with a much deeper analysis of both martial embodiment and feminist analysis.
For the first, I would recommend the following literature (although there is more recent stuff):
McCaughey, M. (1997). Real knockouts: The physical feminism of women’s self-defence. New York, NY: New York City Press. (I think that McCaughey's work would really help you)
McCaughey, M. (1998). The fighting spirit: Women’s self-defense training and the discourse of sexed embodiment. Gender and Society, 12(3), 277–300.
Young, I. M. 1998. “Situated Bodies: Throwing Like a Girl.” In Body and Flesh: A Philosophical Reader, edited by D. Welton, 259–273. Oxford: Blackwell. (I specifically added the later Young article because it addresses her earlier article after 25 years of Title IX in the US. I presume there is a similar expansion of sporting opportunities for women in Spain over this period. In this later paper, Young explains the development of an embodiment that is significantly different to that described in Throwing Like A Girl. For Young, embodiment of women is learnt and disciplined, and resistance is possible and growing.)
Thank you again for your suggestions. We inserted some literature that both serves our topic and gives a vaster perspective to readers on martial embodiment, even though our position is not necessarily in agreement with McCaughey's. We believe in the potential of MACS in empowering people, but we also know that they may be conducted in a disempowering, if not traumatic, manner. So, we want to contribute to an expansion of research in martial arts not promoting their myths but advocating for their transformation by showing their problems as well. They may help women, but they may also make women’s lives worse at times. We disagree with McCaughey’s approach of training in spaces that seem to be not safe and may be more traumatic than street experiences. Physical feminism is completely relevant, and we have disused it in the PhD thesis of the first author, but we are careful when speaking about it, making several points due to complexity. We consider that we do not have enough space, in terms of word limit, to enter this topic in this paper, nor is this the focus of the article.
For the second, and without wishing to influence you in ways that you do not want to go, consider the current literature on feminist new materialism, which explains the ways that the body interacts with material ‘reality’ in the spaces it occupies. That is a massive simplification, but I found the following literature useful:
Bargetz, B. and Sanos, S. 2020. Feminist matters, critique and the future of the political. Feminist Theory, 21 (4), 501-516. 10.1177/1464700120967311 (Uses Karen Barad on new feminist materialism)
Simone Fullagar (2017) Post-qualitative inquiry and the new materialist turn: implications for sport, health and physical culture research, Qualitative Research in Sport, Exercise and Health, 9:2, 247-257, DOI: 10.1080/2159676X.2016.1273896 (You mention Simone Fullager’s article in note 7, but her paper offers a great framework for your paper.)
Toffoletti, K., J. Francombe-Webb, and H. Thorpe. 2018. “Femininities, Sport and Physical Culture in Postfeminist, Neoliberal Times.” In New Sporting Femininities: Embodied Politics in Postfeminist Times, edited by K. Toffoletti, H. Thorpe, and J. Francombe-Webb, 1–19. London: Palgrave Macmillan. (Any of Toffoletti’s, Thorpe’s, Olive’s or Pavliis’ recent work would be useful)
Olive, R., and H. Thorpe. 2018. “Feminist Ethnography and Physical Culture: Towards Reflexive, Political and Collaborative Methods.” In Physical Culture, Ethnography and the Body: Theory, Method and Praxis, edited by M. Giardina and M. Donnelly, 114–128. London: Routledge.
Pavlidis, A. 2013. “Writing Resistance in Roller Derby: Making the Case for Auto/Ethnographic Writing in Feminist Leisure Research.” Journal of Leisure Research 45 (5): 661–676. 10.18666/jlr-2013-v45-i5-4368.
van Ingen, C., 2016. Getting lost as a way of knowing: the art of boxing within Shape Your Life. Qualitative research in sport, exercise and health, 8 (5), 472- 486. 10.1080/2159676X.2016.1211170
We have included some of these suggestions, which are highly relevant and make our paper stronger.
If you don’t want to go there, then I think that you need to dive a bit deeper into the literature on the patriarchal context of sport, and the various methods that reinforce the hierarchy that places men over women. You certainly covered this idea when talking about warriors, but this also needed a deeper conceptualization.
One final comment on content- I’d remove footnote 2. It does not add to your paper and potentially detracts from the potential for women to find resistant subjectivities through karate. My very limited understanding of karate is that women often spar with men in training. Additionally, the rigorous educational and accreditation systems (belts) along with the rules of engagement, would allow for the theoretical possibility of mixing the genders in combat. I acknowledge your final point in this note, but someone like McCaughey would probably argue differently. But this may be especially so when you consider what your research said about authenticity and warriors.
Thank you for your comment. We decided to keep the note, though, because this is a current contested terrain among scholars in martial arts. Our advocacy is towards inclusion; however, inclusion is possible in recreational sport. High-level competitive sport is about minimum differences counting to win. As you exemplified sparring between men and women, we follow this line and do not enter other genders for this specific argument. A woman fighting a man in a competition is not a minimum difference. For sparring, in daily dojos, it is perfectly fine, because that is sport “for fun”. Sport “for real”, in competitions, is about playing the rules. So, there will be strong blows displayed under the rules and the material reality of male and female bodies play then a fundamental role. In this case, De Beauvoir mentioning physiology and hormones, as Young (1998) comments, makes a lot of sense. We have a paper in press related to how women high-level competitors do not necessarily feel safe or able to face violence. Following your argument, belts also do not attest that a person is a gun in themselves just because they hold a black belt. Martial arts are also a business and an accepted social place where exerting power over others is praised. Thus, people may be high in the formal hierarchy of the belts, but low in hierarchies for other forms of capitals. This is why, finally, our advocacy is for equity.
One specific comment- delete the sentence (or at least the reference) on p.4 line 186-187 to Bowman. Bowman's paper is about how Covid impacted on his relationship with martial arts, and not about how Covid impacted on research design.
It has been deleted. Thank you for your observation.
Comments on the Quality of English Language
I want to acknowledge the efforts of the author(s) to translate participant comments across languages. I have never had to endure this, and it must be difficult.
Thank you for your acknowledgement. Indeed, English is the first author’s third language, notwithstanding, we really believe and are passionate about our topic, and committed to the proposal of change in MACS environments. Then, even aware of the required effort, we want to make our work visible. We have reviewed the text trying to make it clearer and more objective. Several deleted words cannot be seen, but changes made are highlighted along the paper.
My comments will be mostly critical (that is the nature of a review unfortunately), but I also hope that they are constructive.
I found that the authors used a number of complex sentences where the understanding of the manuscript would be improved with more and shorter sentences and paragraphs. A strong edit to remove wordiness where it is distracting is needed. I suspect that there were two reasons for this:
- A) An endeavour to squeeze this research into the idea of ‘contested terrains’ by mentioning this concept as often as possible- In my view this was both unnecessary and distracting.
e.g. p. 2 line 76-79- I don’t see the need to define contested terrain here. If you want to do so, then shorten to ‘a site of struggle…involving key interest groups with varying resources and material interests.’
We shortened this definition (and other sentences) but specifically here we kept the final part because “beliefs” are relevant to the context that often conceive karate mythologically or even as a sort of religion.
- B) A desire to constantly link back to existing research literature. This again became distracting in trying to follow the narrative flow of the paper.
e.g. p.4 lines 160-165- You could easily cut out the references to Hobsbawn and Williams, and probably Bourdieu and Green, and have a much tighter sentence structure which emphasizes the main point such as:
'The history of, and philosophies underpinning, martial arts create a field where the teachings of a school or master set up hierarchical power relations.'
These are two examples of many in the paper where you could lose words whilst producing a stronger narrative flow.
Thank you again. We “cleaned” the paper following your recommendation, keeping, though, references that are relevant to our argument, e.g., manipulations carried out under the excuse of tradition.
Finally, you need to redo the reference list to use a consistent style.
We have reviewed the references following one single style.
We appreciate your suggestions, have learned from the review process, and are deeply grateful in seeing that even when there is no complete agreement (following, for example, the suggestion on McCaughey), it is still possible to work together and grow. The martial arts world is a structured world, not willing to change, and it seems to be kept in MACS scholarly, not always welcoming critical views.
Reviewer 2 Report
This is a great article, engaging and thoughtful.
My recommendation is for a few minor revisions to strengthen an already strong contribution to the literature on gender and sports.
Strong theoretical framing and analysis of karate narratives. Some further elaboration and clarification in places would strengthen, for example, on embodied subjectivity (a process that does not 'end', that is ongoing - whereas the authors seem to suggest that one 'becomes' a subject).
I have attached a draft of the article with my notes included, but will note a few of the main revision suggestions here.
First, the article would benefit from some historical context on karate in Spain, including how it has been taken up and is practiced differently within Spain (noting forms of Euro appropriation?). The authors note the tension and/or balance negotiated between karate as a martial art and as a sport, but further elaboration and examples of how athletes experience this would be good (think this is tied to discussion on page 6 - that the athletes consider themselves like other athletes, but also with additional special characteristics).
Second, some introduction to the karate habits and its gendered dynamics earlier on in the paper (perhaps on page 2 when habitus is introduced) would be useful and would offer some framing / context for the analysis.
I recommend the authors reflect a bit more on the space that karateka women embody - similar to other female athletes, they are often positioned as negotiating an embodiment (that is constantly shifting) of femininity and masculinity - and what their analysis offers the field of gender and sports. Further there is discussion of how karateka women practice karate differently than men that suggests a disruption of the gender binary (using their head, being calculating, rather than drawing on impulse/drive - emotion) that could be highlighted more. The notion that women athletes have to negotiate gender within sport in various and ongoing ways (similar to all women, but more focused - can't be too strong / act too 'masculine' but also demonstrate that women can embody these characteristics and are not passive or weak) is not new, so I encourage authors to draw out what their analysis of karateka women contributes to the field a bit more. How does karate (habitus) offer new ways to think about the embodiment (and disruption) of normative gender binary, that could be incorporated into other sporting spaces?
Some suggestions for article to refer to that would support analysis are:
(2015) “So Tight in the Thighs, So Loose in the Waist”, Feminist Media Studies, 15:6, 1035-1052, DOI: 10.1080/14680777.2015.1033638
Toffoletti, K., H. Thorpe, and J. Francombe-Webb. 2018. New Sporting Femininities: Embodied Politics in Postfeminist Times. London: Palgrave MacMillan.
Hardy, Elizabeth. 2015. “The Female ‘Apologetic’ Behavior within Canadian Women’s Rugby: Athlete Perceptions and Media Influences.” Sport in Society 18, no. 2: 155–67.
Finally, some terms and/or sentences I suggested editing and maybe due to translation issues. Of note is 'homosexuality' - which may still be a term used in Spain but is over thought of as a medicalized term and therefore not used in North America (queer or LGBT in its place). I have highlighted some suggested words or phrases, but leave it to the author to leave or change as they see fit.
As I imagine is obvious, I thoroughly enjoyed the article and am grateful for the opportunity to review it.

Author Response
Please see below my replies in green. Thank you.
Reviewer 2
This is a great article, engaging and thoughtful.
My recommendation is for a few minor revisions to strengthen an already strong contribution to the literature on gender and sports.
Thank you very much for your kind words.
Strong theoretical framing and analysis of karate narratives. Some further elaboration and clarification in places would strengthen, for example, on embodied subjectivity (a process that does not 'end', that is ongoing - whereas the authors seem to suggest that one 'becomes' a subject).
Thank you for your observation. We have inserted content in the first paragraph of the session on embodied subjectivity to address the point you made.
I have attached a draft of the article with my notes included, but will note a few of the main revision suggestions here.
That was very helpful. We have addressed any of your notes as you can see highlighted in the submitted version of the manuscript.
First, the article would benefit from some historical context on karate in Spain, including how it has been taken up and is practiced differently within Spain (noting forms of Euro appropriation?). The authors note the tension and/or balance negotiated between karate as a martial art and as a sport, but further elaboration and examples of how athletes experience this would be good (think this is tied to discussion on page 6 - that the athletes consider themselves like other athletes, but also with additional special characteristics).
Thank you for your point. This paper is part of a thesis in which the first author dedicated themselves to extensively describing karate since its origins. We have been publishing other articles where we delve into such historical aspects, and we do not intend to focus on them in this specific paper. You asked on page 4 about who invented the martial tradition and we understand you are referring to this historical aspect as well. Notwithstanding, we do not explain it in detail because the literature we are using already says that. History, and traditions, is/are told by those who hold the means for doing so and they do it in a way that beneficiates them, selecting parts according to convenience.
Second, some introduction to the karate habits and its gendered dynamics earlier on in the paper (perhaps on page 2 when habitus is introduced) would be useful and would offer some framing / context for the analysis.
I recommend the authors reflect a bit more on the space that karateka women embody - similar to other female athletes, they are often positioned as negotiating an embodiment (that is constantly shifting) of femininity and masculinity - and what their analysis offers the field of gender and sports. Further there is discussion of how karateka women practice karate differently than men that suggests a disruption of the gender binary (using their head, being calculating, rather than drawing on impulse/drive - emotion) that could be highlighted more. The notion that women athletes have to negotiate gender within sport in various and ongoing ways (similar to all women, but more focused - can't be too strong / act too 'masculine' but also demonstrate that women can embody these characteristics and are not passive or weak) is not new, so I encourage authors to draw out what their analysis of karateka women contributes to the field a bit more. How does karate (habitus) offer new ways to think about the embodiment (and disruption) of normative gender binary, that could be incorporated into other sporting spaces?
Some suggestions for article to refer to that would support analysis are:
Dawn Heinecken (2015) “So Tight in the Thighs, So Loose in the Waist”, Feminist Media Studies, 15:6, 1035-1052, DOI: 10.1080/14680777.2015.1033638
Toffoletti, K., H. Thorpe, and J. Francombe-Webb. 2018. New Sporting Femininities: Embodied Politics in Postfeminist Times. London: Palgrave MacMillan.
Hardy, Elizabeth. 2015. “The Female ‘Apologetic’ Behavior within Canadian Women’s Rugby: Athlete Perceptions and Media Influences.” Sport in Society 18, no. 2: 155–67.
Thank you for your suggestions. We inserted some literature that both serves our topic and gives a vaster perspective to readers on martial embodiment.
Finally, some terms and/or sentences I suggested editing and maybe due to translation issues. Of note is 'homosexuality' - which may still be a term used in Spain but is over thought of as a medicalized term and therefore not used in North America (queer or LGBT in its place). I have highlighted some suggested words or phrases, but leave it to the author to leave or change as they see fit.
We have reviewed the text trying to make it clearer and more objective. Several deleted words cannot be seen, but changes made are highlighted along the paper. Thank you for your observation on the term “homosexuality”. Spain indeed still accepts its use, but we prefer conforming with a more updated and well-employed language because words do matter.
As I imagine is obvious, I thoroughly enjoyed the article and am grateful for the opportunity to review it.
We appreciate your careful review and all notes through the PDF. We have learned from this process and believe that our article is stronger now. Thank you!

Round 2
Reviewer 1 Report
Thanks for assessing all comments made in the previous review. I realize that most of my comments were about 'what the authors could do' and were impossible to achieve given word length restrictions. I note that you have addressed as many suggestions as you could.
I also acknowledge your discussion of the limitations of McCaughey's position, and why the context of your paper may not fit with this position.